

# SOC stabilization mechanisms and temperature sensitivity in old terraced soils

Pengzhi Zhao[1], Daniel J. Fallu[2], Sara Cucchiaro[3,4], Paolo Tarolli[3], Clive Waddington[5], David Cockcroft[5], Lisa Snape[6], Andreas Lang[6], Sebastian Doetterl[7], Antony G. Brown[2,8], Kristof Van Oost[1]

[1]Georges Lemaître Centre for Earth and Climate Research, Earth and Life Institute, UCLouvain, 1348 Louvain-la-Neuve, Belgium

[2]Tromso University Museum, UiT The Artic University of Norway, Kvaløyen 30, 9013 Tromsø, Norway

[3]Department of Agricultural, Food, Environmental and Animal Sciences, University of Udine, Via delle Scienze, 206, 33100 Udine, Italy

[4]Department of Land, Environment, Agriculture and Forestry, University of Padova, viale dell'Università 16, 35020 Legnaro, Italy

[5]Archaeological Research Services, Ltd, Angel House, Portland Square, Bakewell DE45 1HB, UK

[6]Department of Geography and Geology, University of Salzburg, Salzburg, 5020, Austria

[7]Department of Environmental Systems Science, ETH Zurich, Universitätstrasse 16, 8092 Zürich, Switzerland

[8]Geography and Environmental Science, University of Southampton, Highfield SO17 1BJ, Southampton, UK

*Correspondence to*: P. Zhao (penzhi.zhao@uclouvain.be)

**Abstract.** Being the most common and widest spread man-made landform, terrace construction has resulted in an extensive perturbation of the land surface. Our mechanistic understanding of soil organic carbon (SOC) (de-) stabilization mechanisms and of the persistence of SOC stored in terraced soils, however, is far from complete. Here we explored the factors controlling SOC stability and temperature sensitivity ($Q_{10}$) of abandoned prehistoric agricultural terrace soils in NE England, using soil fractionation and temperature sensitive incubation in combination with measurements of terrace soil burial age. Results showed that although buried terrace soils contained 1.7 times more unprotected SOC (i.e., coarse particulate organic carbon) than non-terraced soils at comparable soil depths, a significantly lower potential soil respiration was observed, relative to a control (non-terraced) profile. This suggests that burial of former topsoil due to terracing provided a mechanism for enhanced C stabilization. Furthermore, we observed a shift in SOC fraction composition from particulate organic C towards mineral protected C with increasing burial age. This clear shift to more processed recalcitrant SOC with soil burial age also contributes to SOC stability in terraced soils. Temperature sensitivity incubations revealed that the dominant controls on $Q_{10}$ depend on the terrace soil burial age. At relatively younger ages of soil burial, the reduction of substrate availability due to SOC mineral protection with ageing attenuates the intrinsic $Q_{10}$ of SOC decomposition. However, as terrace soil becomes older, SOC stocks in deep buried horizons are characterized by a higher temperature sensitivity, potentially resulting from the poor SOC quality





(i.e., soil C:N ratio). In conclusion, terracing in our study site has stabilized SOC as a result of soil burial during terrace construction. The depth-age patterns of $Q_{10}$ and SOC fraction composition of terraced soils observed in our study site differ from those seen in non-terraced soils and this has implications when assessing the effects of climate warming or terrace abandonment on the terrestrial C cycle.

## 1 Introduction

Since post-Neolithic times, the construction of terraces has played an important role in the expansion and intensification of agriculture to meet food production (Brown et al., 2021). Terracing is recognized as a major adaptive strategy for land use in hilly areas and as an efficient conservation practice that provides multiple ecosystem services, e.g., erosion control and enhanced biomass yields, soil water recharge and nutrient storage. (Dunjó et al., 2003; Tarolli et al., 2014, 2015; Wei et al., 2016). At the same time the construction of terraces influences soil carbon (C) dynamics by land-use change and the alteration of local topography and hydrological conditions (Shi et al., 2019; Stavi et al., 2019). Terracing typically creates soils enriched in nutrients and soil organic C (SOC) with a high potential to sequester SOC, relative to non-terraced sloping land. The impact of terracing on SOC stocks has been studied in diverse regions (e.g., Europe (Curtaz et al., 2015; Dunjó et al., 2003; Walter et al., 2003), Asia (Chen et al., 2020; De Blécourt et al., 2014; Shi et al., 2019), South America (Antle et al., 2007) and Africa (Kagabo et al., 2013)); in general, terracing was shown to increase SOC stocks, relative to non-terraced landscapes. However, the underlying SOC stabilization mechanisms in terraced soil systems remain poorly studied and quantified. This is an important knowledge gap as it limits our ability to evaluate how SOC stores in terraced systems and buried soils will respond to present and future perturbations such as terrace degradation or land use change.

Three key stabilization mechanisms that can protect SOC from being mineralized and decomposed in soils have been identified; (i) physical protection, whereby soil aggregates physically protect SOC by isolating microbes and fauna from their substrates and inhibiting the rate of diffusion of oxygen and enzymes (Six et al., 2002). (ii) Chemical stabilization, whereby SOC becomes chemically inaccessible to decomposers resulting from the absorption of SOC onto mineral surfaces or from complexation of dissolved organics with biochemical weathering products such as Fe-, Mn- and Al-oxy-hydroxides (Blume et al., 2016). (iii) Biochemical stabilization, referring to inherent biochemical recalcitrance of SOC. Historically, it was thought that biochemical stabilization was mostly a function of molecular complexity structure of SOC (Lützow et al., 2006). However, analytical and experimental advances have questioned the selective preservation of organic structures due to recalcitrance (Kleber, 2010; Schmidt et al., 2011). Instead, physical and especially chemical mineral related stabilization mechanisms are now considered as the primary mechanisms (Schmidt et al., 2011; Kleber et al. 2021).

Terraces are developed through cut or fill process or as a consequence of active erosion, deposition and cultivation (Mesfin, 2016). Soil redistribution during terrace development results in an exposure of the subsurface soil at the cut or eroding section, and burial of the original topsoil at the fill or deposition section. Thus, the impacts of terrace construction on SOC dynamics





can be compared to the mechanisms proposed for erosional impacts, given the similar soil redistribution patterns, i.e., topsoil removal at erosional position (same as cut section) and soil burial at depositional position (fill section) (De Blécourt et al., 2014; Van Oost et al., 2007). Based on the knowledge from erosional studies, three possible mechanisms that link terracing

with SOC stabilization, acting together or separately, should be considered: (i) the removal of topsoil and physical breakdown of soil aggregates during terrace construction enhances the decomposition of SOC (Bailey et al., 2019; Doetterl et al., 2016; Gao et al., 2020); (ii) SOC removal at the cut or eroding section that is gradually replaced through continued SOC inputs from new photosynthate and plant litter decomposition (Berhe et al., 2007; Harden et al., 1999); (iii) burial of the original topsoil at the fill or deposition section which results in reduced SOC decomposition rates by changing the environmental context for

SOC decomposition, e.g., providing a low-mineralization context by reducing oxygen availability in soil pore space and increasing soil water content (Berhe et al., 2007; Vandenbygaart et al., 2012; Wang et al., 2014; Wiaux et al., 2015). Studies have shown that the stability of buried SOC largely depends on the burial history (Van Oost et al., 2012; Wang et al., 2015).

Both the current SOC status of terraces and the extent to which terraced soil systems will represent a net SOC loss or sink largely depend on the magnitude by which the three mechanisms identified above interact and govern SOC dynamics. This is

of particular importance since predicting the sensitivity of SOC decomposition to temperature change is vital to evaluate the warming-induced changes in soil SOC stocks (Kirschbaum, 1995; Knorr et al., 2005). However, the fundamental drivers and basic patterns of the temperature sensitivity of SOC decomposition in agricultural terraces are poorly described so far. A recent study reported the effect of terrace construction on soil $CO_2$ emission and temperature sensitivity in Chinese Loess Plateau, however, the simulation of topsoil removal during terrace construction only reflected short-term (<2 years) response and

mechanisms (Gao et al., 2020).

On the basis of the fundamental principles of enzyme kinetics, temperature sensitivity of SOC decomposition ($Q_{10}$) should be determined by the chemical complexity of SOC molecules and by the availability of SOC substrate to decomposers (Davidson and Janssens, 2006). On the one hand, chemically more complex SOC molecules (thus SOC of low quality), that is, those that require a high activation energy to degrade, should have a higher $Q_{10}$ than SOC of higher substrate quality (Bosatta and Ågren,

1999). On the other hand, environmental factors (e.g., SOC protection) as a constraint on substrate availability should have a negative effect on observed $Q_{10}$ values (Gillabel et al., 2010). According to the fundamental principles mentioned above, the buried SOC stock at fill sections of a terrace is likely to be less sensitive to warming, given the high quality of this C stock. However, the labile fraction of this buried SOC stock might be gradually decomposed or transferred into protected SOC fractions as the terrace system ages (Wang et al., 2015). This changes the SOC quality and protection (and hence substrate

availability) over time. This time-dependent change in SOC quality and SOC protection may have important implications for the temperature sensitivity of terrace SOC and its feedbacks to climate change.

Here, we present a study carried out on the Plantation Camp prehistoric agricultural terraces in the Ingram Valley of NE England, aiming at improving the mechanistic understanding of SOC dynamics in agricultural terraces and its potential feedback to climate change. To achieve this, we combined SOC fractionation experiments and temperature sensitivity





incubations with measurements of relative terrace soil burial age based on portable optically stimulated luminescence (pOSL) data. The detailed objectives of this study are to investigate (i) the factors controlling SOC stabilization in agricultural terraces and (ii) the associated patterns of SOC temperature sensitivity.

## 2 Material and methods

### 2.1 Study sites

The study area is located in the Ingram Valley (Fig. 1a) in the Cheviot Hills of NE England within the Northumberland National Park (55° 26' 26.14" N, 1° 59'4 9.52" W). This region is characterized by Maritime temperate climate with an average monthly temperature that ranges between -1 °C (in February) and 18 °C (in July and August) and an average annual rainfall of around 650 mm (www.meteoblue.com). Umbrisols have developed from Andesite igneous bedrock of the Cheviot Volcanic Formation formed approximately 393 to 419 million years ago in the Devonian (British Geological Survey. 2018). The study site complex

is one of the largest monuments in England (5.7 km$^2$) and is known as a multi-period archeological landscape and currently also being investigated as part of project TerrACE (https://www.terrace.no/). Previous archaeological investigations during 1997-1999 (Frodsham and Waddington, 2004) indicate that the cultivation terraces have a prehistoric origin and date back as far as the early Bronze Age, c. 1800-1500 BC. The landscape at the time the terraces were built was likely dominated by dwarf shrubs (Ericeacea) and grasses, while during the Romano-British period cereals were identified in a pit feature cut into one of

the terrace surfaces (Frodsham and Waddington, 2004). The study area is covered by terraces in a small area of c. 9000 m$^2$ (Fig. 1c).

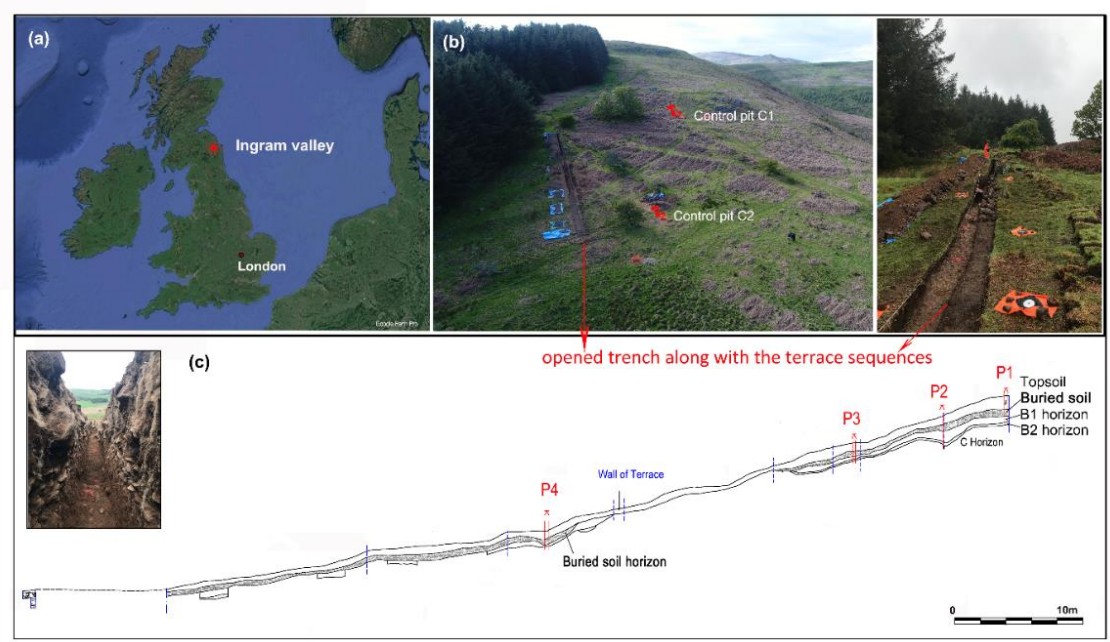

**Fig. 1** Map of (a) studying area (from © Google Maps), (b) sampling site and the (c) excavated terrace trench and sampling positions.






## 2.2 Soil sampling

Topographic data in the sampling area was collected through structure-from-motion (SfM) photogrammetry technique integrating ground-based and UAV (nadir and oblique) images for a complete and detailed survey of terrace system (Cucchiaro et al, 2020a). This technique allowed for the generation of a high-resolution digital terrain models at 0.25–0.10 m resolution,

that helped identify the extent and shape of seven prehistoric agricultural terraces. Moreover, the output of the SfM workflows as point clouds allowed for the extraction of profiles and, sections of the opened trench (Cucchiaro et al., 2020b). A detailed stretch map was created (Fig. 1c; Cockcroft and Waddington, unpublished technical report) for supporting data interpretation. In order to produce a continuous section through seven cultivation terraces, a trench measuring 63 m long by 1.5 m wide was excavated in May 2019 (Fig. 1b). The trench was then carefully cleaned to the depth of the weathered bedrock surface in order

to expose the full nature and extent of any archaeological features, structures and deposits where they survive. Four terraces (P1, P2, P3, P4) were sampled in detail along the trench in order to cover the full range of potential burial ages present in the sequence (Fig 1). Additionally, two control soil pits (C1, C2) were excavated in proximity to the terraces to provide an undisturbed soil profile, i.e., a soil profile that was not subjected to terracing. Soil samples were collected every 5 cm except for P1 where a different depth interval was used for subsoil layers (Table 1). Thus, while our sampling approach only covers

one field trench, our approach is a suitable compromise between the need for spatial representativeness of soil data and the logistical criteria for detailed geo-archaeological excavations at selected sites. The number of highly depth-resolved samples collected this way does allow for the analysis of continuous trends of specific soil parameters within terraced and non-terraced profiles along a sequence of spatially and temporally clearly distinct agricultural terraces. A total of 91 depth-explicit soil samples has been collected this way along the terrace sequence and additional 23 depth-explicit samples collected from the

control pits for further analysis.





**Table 1** Depth layers selected for soil analysis from sampled terraces

| Sampling positions | Sampling depth/cm | Bulk density /cm | SOC fractionation Soil incubation/cm | Total SOC (Depth interval) | pXRF/pOSL (Depth interval) |
|---|---|---|---|---|---|
| Control C1 | 70 | 0-10; 30-40; 50-60 | 0-10; 30-45 60-70 | 5 cm | 5 cm |
| Control C2 | 45 | 0-10; 30-40 | 0-10; 30-45 | 5 cm | 5 cm |
| Terrace P1 | 195 | 0-10 | 0-10; 50-70; 90-100; 140-150 | 0-30 cm: 5 cm; 30-50 cm: 15 cm >50 cm: 10 cm | 5 cm |
| Terrace P2 | 150 | 0-10; 30-40; 60-70 | 0-10; 30-45; 60-70 80-90 | 5 cm | 10 cm |
| Terrace P3 | 80 | 0-10; 30-40 | 0-10; 30-45; 60-70 | 5 cm | 5 cm |
| Terrace P4 | 105 | 0-10; 30-40 60-70 | 0-10; 30-45; 60-70 | 5 cm | 5 cm |

## 2.3 SOC fractionation

Forty gram of 5 mm sieved air-dried soils were fractionated in duplicate to obtain SOC fractions for selected soil layers that represent different layers within each undisturbed and terrace soil (topsoil 0-10 cm, shallow subsoil 30-45 cm, deeper subsoil

$\geq$ 60-70 cm). These depth intervals were chosen to maximize the variation in terrace burial age, which we will use to investigate its effect on C dynamics. We used a fractionation scheme based on the conceptual SOC model proposed by Six et al. (2002, 1998) applying a simplified version of its fractionation protocol (Doetterl et al., 2015). Total SOC was fractionated into coarse particulate organic carbon (cPOM, > 250 μm), microaggregate-associated SOC (M, 250–53 μm) and non-aggregated silt and clay SOC (S+C, < 53 μm). In brief, the microaggregate isolator was placed on a reciprocal shaker and 20 g of air-dried soils

together with 50 glass beads were added on top of a sieve with 250 μm mesh size. Then, the isolator and sieve were flushed continuously with deionized water while shaking at low (= 150 rpm) speed for up to 5 minutes until water flowing out of the device onto a 53 μm mesh size sieve was clear and all aggregates on top of the 250 μm mesh size sieve were broken up. Materials left on the 250 μm mesh size sieve were interpreted as cPOM plus sand. Materials left on the 53 μm mesh size sieve





were interpreted as microaggregates and all <53 µm particles were interpreted as non-aggregated silt and clay. All fractions
were analyzed for total C and N using a VarioMax CN Analyzer (Elementar GmbH, Germany). Samples showed no reaction
when treated with 10% HCl and were considered free of carbonates. cPOM was interpreted as unprotected SOC and while M
and S+C were interpreted as mineral-protected SOC in our analysis. (Gillabel et al., 2010; Six et al., 2002). Soil C:N ratio was
widely considered as an indicator of SOC quality (Sollins et al., 1996; Wang et al., 2018). SOC with a higher C:N ratio, for
example, derived from litter of boreal forests, was considered a low-quality or recalcitrant substrate (Liu et al., 2017). This
was because soil C:N ratio was closely related to fungal-to bacterial growth ratios and therefore to the microbial C use
efficiency of the substrate (Soares & Rousk, 2019).

## 2.4 Temperature-sensitivity based on soil incubations

Three replicate samples of 30 g 2–mm sieved bulk soils from the selected soil layers were chosen for fractionation (see Table
1) of each soil profile were incubated at two different temperature levels (20 and 30 °C) using 380 ml sealed jars. A 10–day
pre-incubation was carried out in order to avoid $CO_2$ pulses caused by soil sample preparation (i.e., sieving, drying and
rewetting), then respiration was monitored during 8 weeks while keeping moisture (60% of soil water holding capacity)
constant during the whole experiment, by periodically adding demineralized water to the samples. The temperature and
moisture level used in the incubation were chosen to provide optimal conditions for microbe activity thus inducing the specific
potential maximum heterotrophic respiration (SPR) (Paul et al., 2001). Then, respiration data were collected to calculate
heterotrophic potential specific soil respiration, expressed as $CO_2$-C per unit mass of soil C. For this, every seven days
throughout the experiment gas samples within the incubation jars were circulated through a Li-830 $CO_2$ Gas Analyzer (LI-
COR, Inc., Netherlands) for determining $CO_2$ concentration. $CO_2$ production was analyzed as the average SPR over the whole
incubation period (pre-incubation excluded). To avoid $CO_2$ saturation effects influencing microbial decomposition processes,
the incubation jars were flushed with fresh air after each measurement and left open between measurement cycles with jars
covered by parafilm to allow for gas exchange while limiting water loss through evaporation. Temperature sensitivity of SPR
($Q_{10}$) was calculated as the difference in SPR of the same aliquot samples at 20 °C to 30 °C incubation temperature (Doetterl
et al., 2018; Paul et al., 2001). The average uncertainty for SPR measurement was 0.5 µgC h$^{-1}$ gSOC$^{-1}$.

## 2.5 Soil burial age and soil geochemical properties along terraces

The main total elemental contents (Fe, Al, Mn, Sr, Rb) of the bulk soil were measured using an Olympus Vanta M series
portable energy dispersive X-ray fluorescence spectrometry (pXRF) at each 5 cm depth for all terrace profiles (Table 1). The
Rb/Sr ratio was applied as a proxy for weathering intensity since both elements fractionate during the weathering processes
due to their different chemical behaviour. Because of the relative inertness of Rb compared to Sr, a higher ratio of Rb/Sr
indicates a higher degree of soil weathering and higher age (An et al., 2018). Soil pH was determined with a soil/solution ratio
of 1:2.5 (w/v) with 0.01 M CaCl$_2$ solution using a pH–meter (Mettler Toledo MP220, Mettler–Toledo, Switzerland). Soil



texture was analyzed using a laser diffraction particle size analyzer (Model LS 13 320; Beckman Coulter Inc., Fullerton, USA) after ultrasonic dispersion and removal of organic matter using $H_2O_2$ (35%) (Beuselinck et al., 1998).

Optically stimulated luminescence (OSL) technique has been widely applied for sediment dating (Brown, et al, 2021). To establish a chronology of soil burial along the terrace sequence, OSL data was obtained using a portable OSL reader. Portable OSL technique was used since it has successfully and frequently been applied in similar settings (e.g., Muñoz-Salinas et al.,
2011; Portenga et al., 2016; Porat et al., 2019). When compared to conventional OSL dating, portable OSL is more efficient in terms of time, cost and labour and can therefore be used for large, untreated samples, especially to clarify deposition processes (Muñoz-Salinas et al., 2011**;** Porat et al., 2019). pOSL was recorded at 5 cm intervals throughout all terrace profiles (P1-P4). Luminescence samples were collected in bulk using blackout bags and a tarp to block out the light. The bags were held against the cleaned section at the sampling location and soil scooped into them, avoiding exposure to the light. Counts
were based on 60 seconds of exposure to blue light using the SUERC Portable OSL Meter. Here, besides others, the signal intensity is a function of mineral composition, content of moisture, and age etc. As mineral composition as well as moisture content was relatively homogenous throughout individual profiles, age is the main discriminant for pOSL activity. The consistently low photon counts at the top of the soil profiles was therefore taken as indicator of very young ages. Thus, pOSL activity was taken as a proxy for the relative terrace soil burial age (following Muñoz-Salinas et al., 2011) where it increased
monotonically with soil depth. Note that the interpretation of relative terrace soil burial age based on pOSL was only applied with depth within individual soil profiles rather than between profiles, to rule out batch effects caused by the drying out of the samples which could not be processed in the field. Where pOSL did not increase consistently with depth, it was considered to indicate soil heterogeneity and mixing. Lastly, soil samples were classified into relatively younger and older soils based on a (visual) change in the soil depth-age relation (see Fig. 2). Then, the relationship of SOC fraction abundance and $Q_{10}$ to soil
burial age were analyzed by comparing sample groups along the four investigated terraces.  The pOSL signal showed that the terrace soil profiles gradually become older towards the bottom (Fig. 2), suggesting that the terraces are most likely developed by stone lines which gradually catch soil material moving downslope, resulting in the former "topsoil" layers were buried in deep soil layers. Therefore, the soil burial age was supposed to play a dominant role in terracing SOC dynamics.







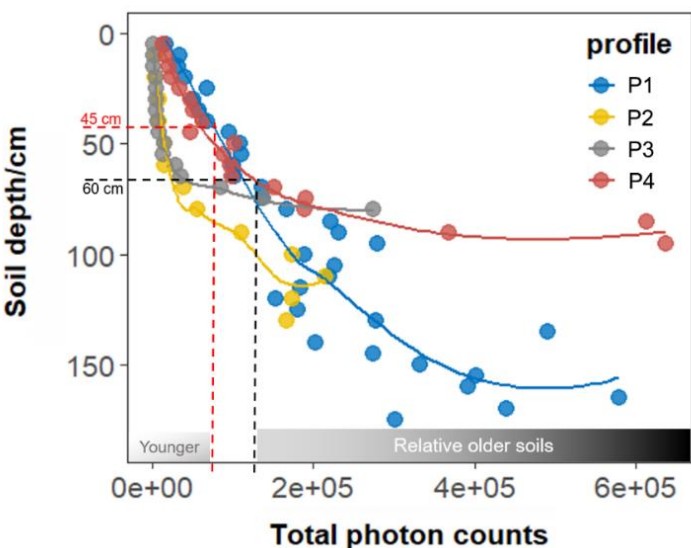

**Fig. 2** The total photon counts versus soil depth in terrace soils. The two soil burial age categories were shown with dashed lines (i.e., above 45 cm = young, below 60 cm = old).

## 2.6 Statistical analysis

All statistical analysis was performed by R 3.6.3 (R Development Core Team, http://www.R–project.org). The differences of SOC concentration between control and terraced soil layers were analyzed with unpaired t-tests. Model fitting function of 'stat_fit_glance' from 'ggpmisc' R package was used to derive non-linear regression model between terrace soil burial age and SOC variables ($Q_{10}$, SPR, SOC fractions). Linear regression and simple correlation were used to examine the relationship between SOC fractions, $Q_{10}$, pH and soil texture. Differences in means of SPR and $Q_{10}$ between the buried and non-buried soil

layers were tested by unpaired t-tests. The 'outlierTest' function from 'car' package was used to identify outliers ($P<0.05$) based on a given model (i.e. cPOM% ~ $Q_{10}$; S+C% ~ SPR; total photon counts ~ $\log(Q_{10})$; total photon counts ~ $\log(SPR)$). Observation for the $Q_{10}$ from P1 topsoil (see gray point in Fig. 7) was identified as an outlier and was therefore not included in the correlation and regression analysis. Significance in all cases was assessed at $P<0.05$. All figures were produced by R package 'ggplot2'.

## 3 Results


### 3.1 SOC fractions composition, temperature sensitivity and soil respiration

For the 0-50 cm soil layers, no significant difference in SOC concentrations could be observed between the terrace soil layers and control soil layers (Fig. 3). Below 50 cm, soil layers in control profiles showed significant lower SOC contents relative to



the terrace soil layers ($P<0.05$). The SOC contents (Fig. S1) of P1 for 95-125 cm, P2 for 65-85 cm, P3 for 50-65 cm and P4

for 55-70 cm depth layers were higher than at the control profiles (C). In combination with the high degree of chemical weathering, we interpreted these layers as buried A horizons (detailed in support information S1).

Considering all profiles, topsoil SOC was mainly composed of C associated with the M fraction, followed by cPOM and S+C (Fig. 4). For the 30-45 cm depth layer, terrace soil layers contained twice as much cPOM associated C than the corresponding soil layer in the control profiles. Below 60 cm (i.e., the buried horizons), the contribution of cPOM to total SOC in terrace soil

layers was 1.7 times higher than in soil layers of the control, while a comparable proportion of physical protected C (M%) was found in terrace and control soil layers. Soil potential respiration (SPR) and SOC temperature sensitivity ($Q_{10}$) varied significantly across soil layers. On average, buried soil layers showed a significant lower SPR than the non-buried layers ($P<0.05$, Fig. 5). While for $Q_{10}$ no significant difference between buried and non-buried soil layers could be found.

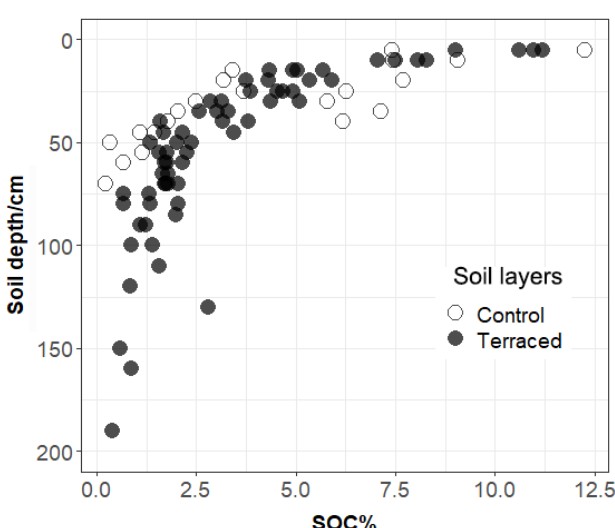

**Fig. 3** Depth profile of SOC concentration. The significant difference ($P<0.05$) in SOC concentration between the terraced and control soil layers can be found below 50 cm layers. No such significant difference at 0-50 cm layers.



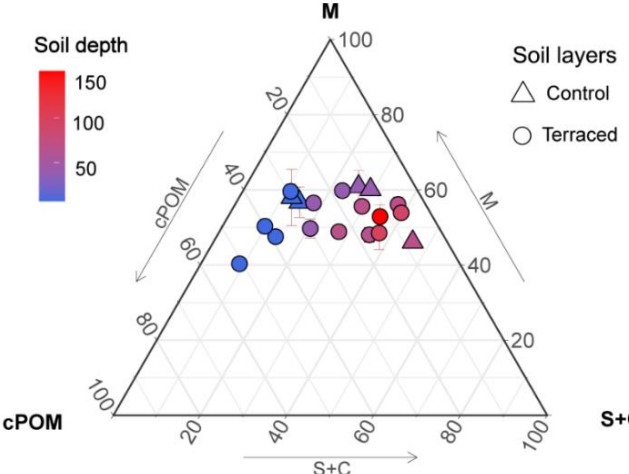

**Fig. 4** Ternary plot of SOC fractions in percentage of total SOC. M= microaggregate associated SOC; cPOM= coarse particulate SOC; S+C=silt & clay associated SOC. Color bar represents the soil depth (cm). Error bars denote one standard deviations of the M fraction measurements (N=2).

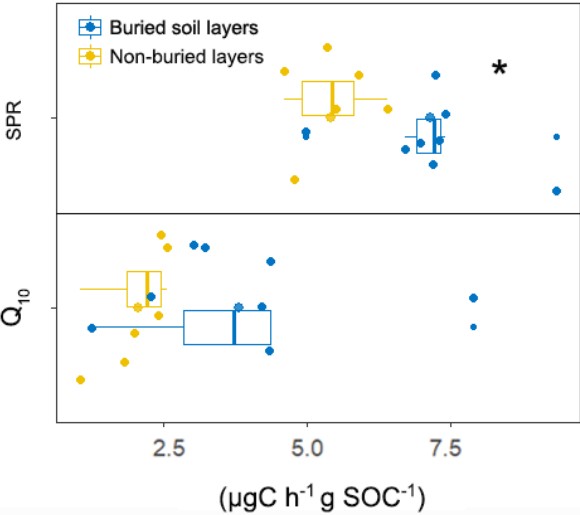

**Fig. 5** Boxplot of soil potential respiration rates (SPR) and SOC temperature sensitivity ($Q_{10}$). * means the significant differences in SPR or $Q_{10}$ between buried and non-buried soil layers ($P<0.05$). Identification of buried soil layers is based on the depth profile of SOC concentration and soil chemical weathering degree with support from the stretch map in Fig. 1 (see details in Support information S1).

### 3.2 Controls on terrace SOC stability

A non-linear regression analysis was used to identify the effect of total photon counts, which treated as a proxy for soil burial 305 age, on SOC fractions distributions and on SPR. We found that SPR was negatively related to total photon counts (Fig. 6a), indicating that the terrace soil burial age was an important control on SPR. Furthermore, cPOM and S+C fractions were significantly correlated with burial age (Fig. 6b), where a shift from cPOM to S+C associated SOC fractions was observed with increasing age. No significant relationship between SPR and measured soil geochemical properties (e.g., pH, soil texture and soil elements) was found (Table 2, $P<0.05$).





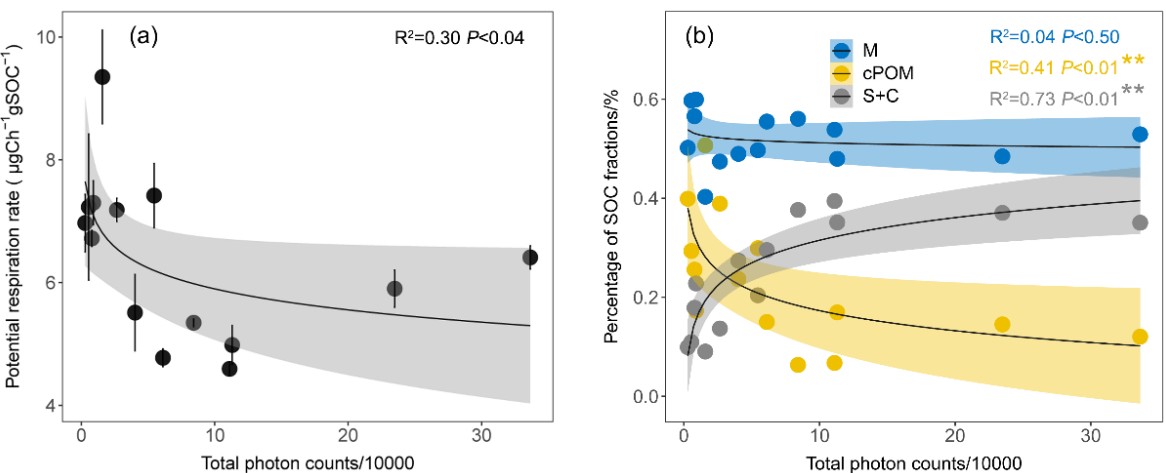

**Fig. 6** Relation between relative terrace soil burial age (total photon counts) and, (a) soil potential respiration rates (SPR) and (b) contribution of SOC fraction to total SOC (%). Error bars denote one standard deviation of SPR measurements (N=3). Formula: y=log(x). cPOM = coarse particulate C; M= microaggregate associated C; S+C=silt & clay associated C. Datapoints are from all terrace profiles. *= $P<0.05$; **$P<0.01$.

**Table 2** Relationships between soil potential respiration (SPR), SOC temperature sensitivity ($Q_{10}$) and measured soil geochemical properties.

|  | pH | Clay | Silt | Sand | Fe | Mn | Al | Rb/Sr |
|---|---|---|---|---|---|---|---|---|
| Relative younger layers | | | | | | | | |
| $Q_{10}$ | -0.68 | -0.90* | 0.35 | 0.31 | -0.75 | -0.55 | -0.56 | -0.14 |
| SPR | -0.17 | -0.40 | -0.15 | 0.43 | -0.09 | -0.45 | -0.38 | 0.72 |
| Relative older layers | | | | | | | | |
| $Q_{10}$ | -0.44 | -0.69 | -0.79* | 0.80* | 0.66 | -0.29 | 0.49 | 0.20 |
| SPR | 0.05 | 0.17 | -0.28 | 0.17 | -0.10 | 0.18 | -0.32 | 0.66 |

* $P<0.05$.

### 3.3 Terrace soil burial age and SOC temperature sensitivity ($Q_{10}$)

A significant relationship between $Q_{10}$ and total photon counts was observed (Fig. 7). This relationship suggests that $Q_{10}$ values decline rapidly with the increasing age for young soil horizons, while for older soil horizons $Q_{10}$ remained relatively constant or increased in oldest soil horizons. Linear regression further showed that for relative younger soil horizons (Fig. 2), $Q_{10}$ values were significantly related to the proportion of S+C associated C and cPOM associated C, while for old soil horizons no such strong relation was found (Fig. 8).





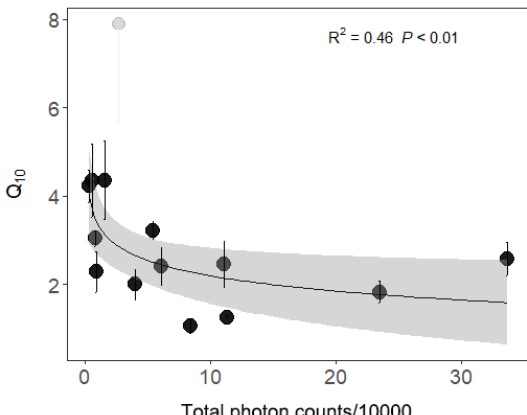

**Fig. 7** Relationship between SOC temperature sensitivity to decomposition ($Q_{10}$) and relative terrace soil burial age (total photo counts). Error bars denote one standard deviation of the $Q_{10}$ measurements (N=3). Formula: y=log(x).

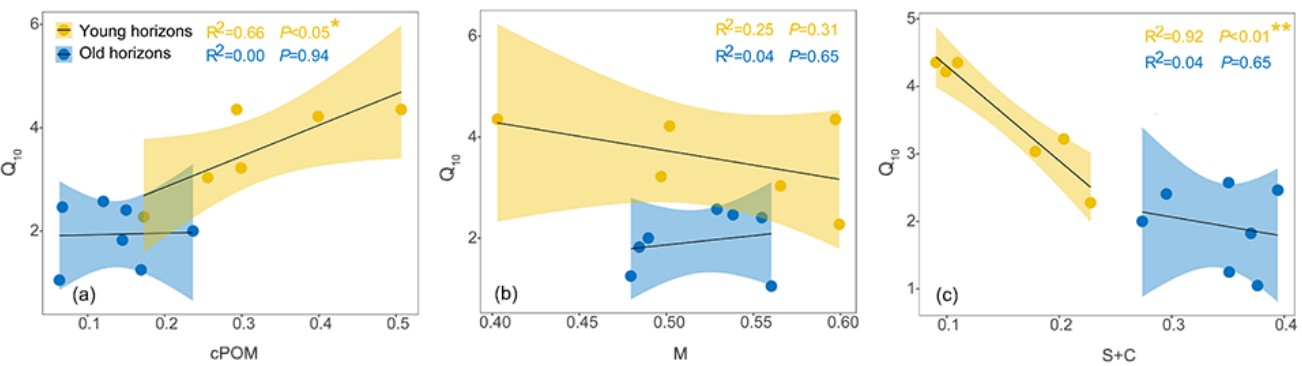

**Fig. 8** Relationship between SOC temperature sensitivity ($Q_{10}$) and (a) coarse particulate C (cPOM), (b) microaggregate associated SOC (M) and (c) silt & clay associated C (S+C) for relative younger and older terrace soil horizons (Fig. 2), respectively. *= $P<0.05$; **$P<0.01$.

In order to further identify the underlying controls, we linked the patterns of $Q_{10}$ to soil C:N ratio and measured soil geochemical properties (e.g., pH, soil texture). We found that $Q_{10}$ was significantly correlated to the C:N ratio of bulk soil, M and S+C fractions (Table 3). In addition, Fig. 9 shows that the C:N ratio of bulk soil and SOC fractions first decreased with

burial age then significantly increased ($P<0.05$) when the terrace soils became very old. Furthermore, the $Q_{10}$ was significantly correlated to clay content at relatively younger burial ages, while silt and sand contents was significantly correlated to $Q_{10}$ at relatively older burial ages (Table 2, $P<0.05$).






**Table 3** Relationship between SOC temperature sensitivity ($Q_{10}$) and C:N ratios of bulk soil and SOC fractions.

|          | Bulk soil | cPOM | M     | S+C   |
|----------|-----------|------|-------|-------|
| $Q_{10}$ | 0.60*     | 0.03 | 0.61* | 0.62* |

\* $P<0.05$. N=13. cPOM= coarse particulate C; M=microaggregate associated C and S+C= silt and clay associated C.

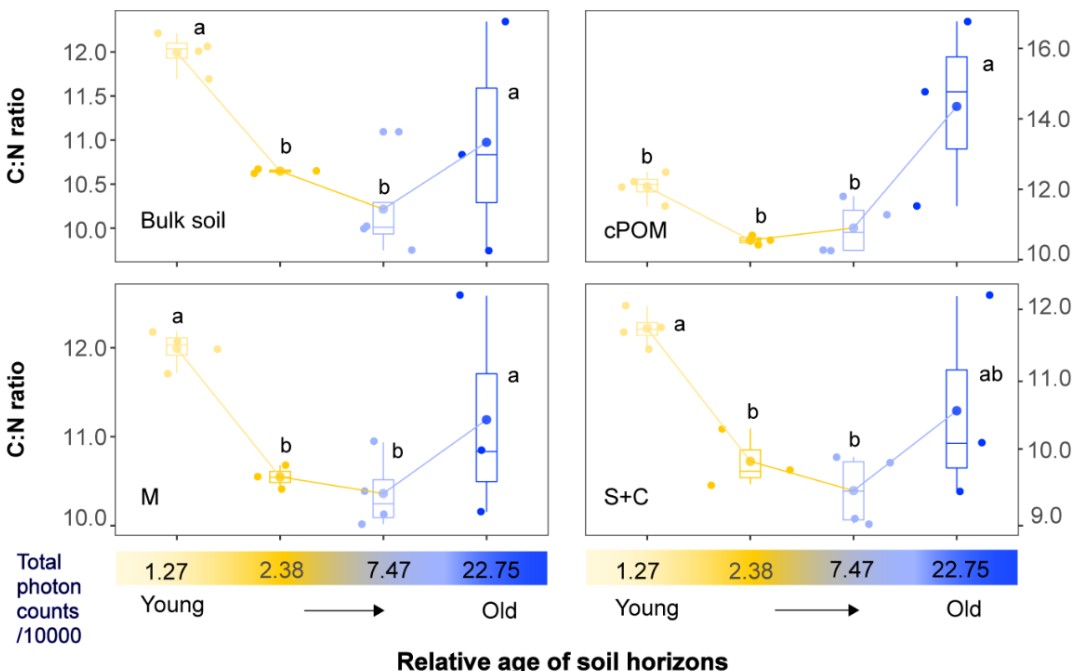

**Fig. 9** Boxplots of C:N ratios for bulk soil and SOC fractions along the gradient of terrace soil burial age (total photon counts). cPOM= coarse particulate C; M=microaggregate associated C and S+C= silt and clay associated C. Significant differences in C:N ratios between soil age gradients were indicated by different lowercase letters ($P<0.05$).

## 4 Discussion

### 4.1 Controls on SOC stability: importance of carbon burial and terrace age

The pOSL signal showed that the terrace soil profiles gradually become older towards the bottom (Fig. 2), suggesting that the terraces are most likely developed by stone lines which gradually catch soil material moving downslope. This process is comparable to the soil redistribution process along catena within eroding landscapes (De Blécourt et al., 2014; Doetterl et al., 2015). Previous work has highlighted that carbon burial due to soil redistribution has important implications for SOC stabilization (e.g., Berhe et al., 2012; VandenBygaart et al., 2012; Van Oost et al., 2012; Wang et al., 2014). Our results show that the buried soil layers (i.e., older soils below 60 cm) have a significant lower SPR than the non-buried layers (Fig. 5),





indicating that carbon burial by terracing reduces soil potential respiration and is therefore a mechanism that contributes to SOC stabilization in agricultural terraces. Although buried A horizons in terrace profiles contain 1.7 times more unprotected SOC (cPOM, Fig. 4) than non-terraced profiles, the less favorable environmental conditions for microbial decomposers (e.g.,

mostly lack of oxygen and water saturation) are likely to constrain in-situ mineralization rates of SOC (De Blécourt et al., 2014; Wang et al., 2014; VandenBygaart et al., 2015). This resulted in a partially-preserved but highly biologically processed SOC store in terraced soils with a lower SPR, relative to non-terraced soils.

The significant relationship between SPR and total photon counts (Fig. 6a) indicates that the time since terracing is a key variable controlling the stability of SOC in terrace soils. More specifically, SPR declines rapidly with increasing terrace soil

burial age until soils become relatively older, leading to an increase of SOC store in the old terraces compared to non-terracd landscape (Chen et al., 2020). In addition, we found that the percentage of unprotected (cPOM) and mineral protected (S+C) SOC fraction is significantly related to the terrace soil burial age (Fig. 6b), indicating a shift from an active SOC fractions with short residence times (higher SPR) at relative younger burial age to a mineral protected SOC fractions with long residence times (lower SPR) at relative older burial age. This clear shift to more processed recalcitrant SOC with age is the underlying

process leading to SOC stabilization in terrace soils. These mechanistic insights provide support for previous studies, reporting that terrace age significantly influences the SOC sequestration benefits of terraces (Chen et al., 2020).

By comparing the cPOM fraction of the topsoil layer (0-10 cm) with previous topsoil (burial horizon, >60 cm), we estimated that about 23-27% of cPOM fraction in buried horizons has been decomposed or transferred into physical protected or mineral protected SOC since the terrace were built (Results 3.1, Fig. 4). Given that the buried soil horizons of terrace profiles still

contain 1.7 times more unprotected SOC (cPOM) than SOC at similar soil depths in control profiles, we suggest that SOC cycling in current terrace has not yet reached a steady state and that the SOC store is most likely to slowly decrease due to cPOM decomposition in the future. We explain this with the observation that the labile fraction of deposited and buried SOC (i.e., cPOM) decomposes more slowly under burial conditions than in topsoil (Fig. 6b; Wang et al., 2014). As a result of the higher amount of unprotected, more readily available C, young terraces (<5 years; e.g., Chen et al., 2020; Gao et al., 2020) are

more likely to be a C source for atmospheric $CO_2$. However, with increasing terrace age, this SOC loss can be gradually balanced out (after 5-29 years, e.g., Chen et al., 2020; De Blécourt et al., 2014) through subsoil minerals stabilizing new OC inputs from roots, dissolved organic carbon and root exudates. This mechanism is promoted in terraces due to the higher net primary productivity of vegetation planted and the greater soil thickness of terraced soils and in line with the concept of dynamic replacement of C in eroding landscape positions (Berhe et al., 2007; Harden et al., 1999; Van Oost et al., 2005; 2007).

**4.2 Main controls on temperature sensitivity**

According to the Arrhenius equation, $Q_{10}$ of SOC decomposition is theoretically jointly determined by the molecular complexity (SOC quality) and availability of the substrates (referred to here as SOC protection) (Davidson and Janssens, 2006). In general, enzymatic decomposition of biochemically recalcitrant substrates (lower SOC quality), require more activation





energy to degrade, and should have a higher temperature sensitivity than the decomposition of more labile substrates (Bosatta
and Ågren, 1999; Craine et al., 2010a). However, environmental constraints (e.g., SOC physical or chemical protection, soil
acidity etc.) can reduce substrate availability, which can dampen the intrinsic temperature sensitivity (Gillabel et al., 2010). It
has been reported that soil pH is an important control on $Q_{10}$ at the landscape scale, by directly affecting microbial biomass,
diversity and, therefore the enzyme activities, or indirectly through altering nutrient solubility, mineral matrix and therefore
the substrate availability (e.g., Craine et al., 2010b; Ali et al., 2018). However, in our study soil pH was not strongly related to
SPR or $Q_{10}$ (Table 2), possibly due to the small variation in soil pH (pH=3.71±0.34). Instead, our results showed that at relative
younger burial ages, $Q_{10}$ is negatively related to SOC mineral protection (Fig. 8) and clay content (Table 2). This indicates that
the adsorption of labile SOC (e.g., dissolved OC) by soil reactive clay minerals forms organo-mineral associations, which
reduce the substrate availability to decomposers (Kögel-Knabner et al., 2008). As a result, SOC mineral protection attenuates
the intrinsic $Q_{10}$ as evidenced by the rapid decrease in $Q_{10}$ with total photo counts at relatively young age stage. However, at
relative older soil burial stages, we could not detect any significantly correlation between $Q_{10}$ and SOC fractions (Fig. 8), while
C:N ratio of bulk soil, M and S+C soil fractions were significantly related to $Q_{10}$ (Table 2). More importantly, we observed a
significant increase in C:N ratio for bulk soil and all SOC fractions at the oldest buried soil layers (Fig. 9). We proposed two
possible reasons as to why C:N ratios were closely linked to $Q_{10}$, especially in oldest buried soil layers. Both depth profile of
soil chemical weathering, SOC concentration and field investigation indicate that these oldest soil horizons (Fig. 9, total photon
counts/1000=22.5) are likely from the former buried topsoil (detailed in support information S1). The former topsoil may still
contain less decomposed organic litter, which is commonly reflected in a higher soil C:N ratio (Xia et al., 2021). Alternatively,
unlike for fresh litter, SOC with a higher C:N ratio is usually considered as lower-quality or recalcitrant substrate, which are
less bioavailable for soil microorganisms (e.g. Sollins et al., 1996; Liu et al., 2017; Wang et al., 2018). The observed increase
in soil C:N ratios at greater depths in older terraces might therefore reflect a shift towards poor quality SOC substrate (Ali et
al., 2018) which explains the observed increase of $Q_{10}$ in the oldest soil horizons (i.e., Fig. 7a, Result 3.3). We propose that the
latter is more likely in our experiments. Together, our results suggest that the factors controlling SOC temperature sensitivity
of terrace soils (SOC mineral protection or C:N ratio) change over time. At relatively younger stages of soil burial, the
reduction of substrate availability due to the increased SOC mineral protection with ageing attenuates the intrinsic $Q_{10}$ of SOC
decomposition (Fig. 7, 8). Whereas at oldest terrace soil burial age, SOC in buried layers reflects an increased $Q_{10}$ probably
due to 1) lower C-quality (higher C:N ratio, Fig. 9) and 2) weaker SOC mineral protection (Fig. 6b).

Combining all findings together, we suggest that, in a first phase, the labile SOC fraction (i.e., cPOM) decomposes rapidly
with increasing terrace soil burial age, while at the same time physical protected SOC and to a larger extent, mineral protected
SOC, are accumulating and hence, SOC protection increases. In the context of temperature sensitivity, this SOC protection
increase is more important than the decrease in SOC quality (as represented by the C:N ratio), and as a result, a decrease in
$Q_{10}$ appears with ageing at relatively young burial ages. However, a second phase, significantly increased C:N ratios (i.e.,
decreased SOC quality) at older burial ages result in an increase of $Q_{10}$ (Fig. 7 and Fig. 9). Hence, the temperature sensitivity





of SOC decomposition in the investigated terrace soils, mediated by burial age, is controlled by SOC mineral protection and the C:N ratio.

## 4 Conclusion

Terracing currently has buried a substantial amount of former topsoil SOC (De Blécourt et al., 2014). Soil burial in agricultural terraces has been identified as an important factor enhancing SOC stabilization due to the less favourable environmental conditions for SOC decomposition. Although about 23-27% of the unprotected SOC fraction (cPOM) in buried horizons was decomposed since terrace development, these horizons contain 1.7 times more unprotected SOC than a non-terraced profile at comparable soil depths. Furthermore, we found that the evolution of SOC fractions along with terrace soil burial age is related

to the depth patterns of specific soil respiration, i.e., there is a shift from an active SOC fractions at relatively younger soil burial ages towards more processed recalcitrant SOC fractions at older soil burial ages.

Furthermore, our study provides empirical evidence for the drivers and basic patterns of the temperature sensitivity of SOC decomposition in agricultural terraces. We found that both the C:N ratio and SOC mineral protection regulate soil $Q_{10}$. However which mechanism predominantly controls soil $Q_{10}$ depends on the age of the buried terrace soils. At relatively

younger ages of soil burial, the reduction of substrate availability due to the increased SOC mineral protection with ageing attenuates the intrinsic $Q_{10}$ of SOC decomposition. Whereas at older terrace soil burial age, higher C:N ratio (poor C-quality) result in an increased $Q_{10}$ in deep horizons. It is expected that the ongoing evolution of SOC fractions and the associated changes in soil C:N ratio due to terrace ageing will slowly but steadily destabilize buried former topsoil SOC.

*Data availability.* All data used and produced in this study will be available through Pangaea. (DOI is not yet assigned)

*Author Contributions.* Conceptualization, P.Z., K.V.O., A.G.B.; terrace excavation and soil sampling, C.W., D.C., D.J.F., P.Z., A.G.B., A.L.; soil analysis, P.Z., D.J.F.; topography survey, S.C. and P.T.; data analysis and interpretation, P.Z., K.V.O., A.G.B., D.J.F., L.S., A.L., S.D.; writing-original draft, P.Z.; writing-review and editing, P.Z., K.V.O., S.D., A.G.B.; project administration, A.G.B. and K.V.O.; supervision, K.V.O.; All authors have read and agreed to the published version of the

manuscript.

*Competing interests.* The authors declare that they have no conflict of interest.

*Acknowledgments.* This research was financially supported by the European Research Council, advanced grant for the TerrACE research project (Grant No. 787790, https://www.terrace.no). P.Z. wish to thank the joint grant from China Scholarship Council and UCLouvain (No. 201706600009). The authors would like to thank the volunteers who help with the

field excavation in Ingram valley, UK. The authors also thank Marco Bravin for his assistance with the SOC sample analysis. K Van Oost is a Research Director of the FNRS Belgium.



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






**Table cations**

**Table 1** Depth layers selected for soil analysis from sampled terraces

**Table 2** Relationships between soil potential respiration (SPR), SOC temperature sensitivity ($Q_{10}$) and measured soil geochemical properties.

**Table 3** Relationship between SOC temperature sensitivity ($Q_{10}$) and C:N ratios of bulk soil and SOC fractions.







**Figure cations**

**Fig. 1** Map of (a) studying area, (b) sampling site and the (c) excavated terrace trench and sampling profiles.

**Fig. 2** The total photon counts versus soil depth in terrace soils. The two soil burial age categories are shown with dashed lines (i.e., above 45 cm = young, below 60 cm = old).

**Fig. 3** Depth profile of SOC concentration. The significant difference ($P<0.05$) in SOC concentration between the terraced and control soil

layers can be found below 50 cm layers. No such significant difference at 0-50 cm layers.

**Fig. 4** Ternary plot of SOC fractions in percentage of total SOC. cPOM= coarse particulate SOC; M= microaggregate associated SOC; S+C=silt & clay associated SOC. Color bar represents the soil depth (cm). Error bars denote one standard deviations of the M fraction measurements (N=2).

**Fig. 5** Boxplot of soil potential respiration rates (SPR) and SOC temperature sensitivity ($Q_{10}$). * means the significant differences in SPR or

$Q_{10}$ between buried and non-buried soil layers ($P<0.05$). Identification of buried soil layers is based on the depth profile of SOC concentration and soil chemical weathering degree with support from the stretch map in Fig. 1 (see details in Support information S1).

**Fig. 6** Relation between relative terrace soil burial age (total photon counts) and, (a) soil potential respiration rates (SPR) and (b) contribution of SOC fraction to total SOC (%). Error bars denote one standard deviation of SPR measurements (N=3). Formula: y=log(x). cPOM = coarse particulate SOC; M= microaggregate associated SOC; S+C=silt & clay associated SOC. Datapoints are from all terrace profiles. *= $P<0.05$;

**$P<0.01$.

**Fig. 7** Relationship between SOC temperature sensitivity to decomposition ($Q_{10}$) and relative terrace soil burial age (total photo counts). Error bars denote one standard deviation of the $Q_{10}$ measurements (N=3). Formula: y=log(x). **$P<0.01$.

**Fig. 8** Relationship between SOC temperature sensitivity ($Q_{10}$) and (a) coarse particulate SOC, (b) microaggregate associated SOC (M) and (c) silt & clay associated SOC (S+C) for relative younger and older terrace soil horizons (Fig. 2), respectively. *= $P<0.05$; **$P<0.01$.

**Fig. 9** Boxplots of C:N ratios for bulk soil and SOC fractions along the gradient of terrace soil burial age (total photon counts). cPOM= coarse particulate SOC; M=microaggregate associated SOC and S+C= silt and clay associated SOC. Significant differences in C:N ratios between soil age gradients are indicated by different lowercase letters ($P<0.05$).