# Peer review of "SOC stabilization mechanisms and temperature sensitivity in old terraced soils"

_Biogeosciences, 2021_

## Referee Comment (RC2)

This manuscript explored the controlling factors for SOC stability and temperature sensitivity of its decomposition in agricultural terrace soils. The result is interesting and has a value for evaluate SOC stability under man-made landform. The manuscript falls into the scope of BG, although the Introduction and Discussion sections need to be improved.

Introduction is generally too long, and should be reorganized. the first paragraph was to show the significance of "SOC stabilization mechanisms and temperature sensitivity in terraced soils", and then What's the factors influenced SOC stabilization mechanisms and temperature sensitivity, …. terraced soils affected can vary which factors….. aim of this study?....what's your hypothesis…and finally add a sentence or two about the research expectations or significance in the end ? Authors excessively cleared the well-known organic dynamic mechanisms, therefore, the relative information should be simplified.

Line 170-174, Delete the sentences "Soil C:N ratio….. of the substrate (Soares &Rousk, 2019)", the information should not appear in the Material and Method section.

Discussion section is unnecessarily long and need thorough revision. The title of "4.1 Controls on SOC stability: importance of carbon burial and terrace age" and "4.2 Main controls on temperature sensitivity" are basically ok, but in each part, the logical thread of the discussion part is not clear.

---

## Author Comment (AC1)

**Response to Anonymous Referee #1**

We gratefully thank the referee for his/her constructive comments and have revised the manuscript accordingly. In our response below, referee comments are shown in black, our response in blue.

**General Comments:**

I have reviewed " SOC stabilization mechanisms and temperature sensitivity in old terraced soils." The study aimed to explore the stabilization mechanisms and temperature response of SOC in an old terraced soil". The topic is very interesting and within the scope of the journal. Generally, the manuscript is very well structured and presents the obtained results. The conclusion has highlighted the implication of terracing on soil carbon stabilization in the context of climate warming. However, below are some corrections:

**Reply**: We gratefully thank the reviewer for this positive evaluation and constructive comments, which have been very helpful to improve the quality of the manuscript.

Lines, 15-20. Our mechanistic understanding of soil organic carbon (SOC) (de-) stabilization mechanisms and of the persistence of SOC stored in terraced soils, however, is far from complete. Here we explored the factors controlling 20 SOC stability and temperature sensitivity (Q10) of abandoned prehistoric agricultural terrace soils in NE England, using soil fractionation and temperature sensitive incubation in combination with measurements of terrace soil burial age. Should be replaced with "However, our mechanistic understanding of soil organic carbon (SOC) (de-) stabilization mechanisms and the persistence of SOC stored in terraced soils is far from complete. Here we explored the factors controlling SOC stability and temperature sensitivity (Q10) of abandoned prehistoric agricultural terrace soils in NE England, using soil fractionation and temperature sensitive incubation combined with terrace soil burial age measurements."

**Reply**: Thanks for the comment. We will revise the sentences accordingly.

Line 20 suggest that burial should be "suggest that the burial" Line 55 function of molecular complexity should be "function of the molecular complexity, through cut should be "through the cut." Line 70 Both the current SOC status should be "The current SOC status"

**Reply**: Thank you for the specificity, this will be changed in the manuscript.

Lines 159 that represent should be replaced with "representing" Lines 255 between the terrace soil layers and control soil layers should be "between the terrace and control soil layers"

**Reply**: We will revise the text accordingly.

Lines 320 relative should be changed to "relatively." Lines 335 In order to further should be "To further"

**Reply**: Agreed, this will be revised in the manuscript.

Lines 360 have a significant lower should be "have a significantly lower."

**Reply**: This will be changed in the manuscript.

Lines 370 non-terraced should be corrected "non-terraced."

**Reply**: Thanks for the comment. We will correct the sentence.

Lines 380 with the observation should be changed to "by observing."

**Reply**: We agree and will correct this sentence.

Lines 390 require should be "requires" Lines 395 which can dampen, consider changed to "dampening."

**Reply**: We agree and will revise this.

Lines 400 relative should be changed to "relatively."

**Reply**: Thanks for the comment. We will correct this throughout the manuscript.

---

## Author Comment (AC2)

**Response to Anonymous Referee #2**

We gratefully thank the referee for his/her constructive comments and have revised the manuscript accordingly. In our response below, referee comments are shown in black, our response in blue.

**General Comments:**

This manuscript explored the controlling factors for SOC stability and temperature sensitivity of its decomposition in agricultural terrace soils. The result is interesting and has a value for evaluate SOC stability under man-made landform. The manuscript falls into the scope of BG, although the Introduction and Discussion sections need to be improved.

**Reply**: We gratefully thank the referee for these positive and constructive comments. We have identified two main suggestions raised by the referee 1) Introduction is too long and need to be reorganized; 2) the logical thread of the discussion part is not so clear. We will re-work our Introduction and Discussion in the revised manuscript as follows:

Introduction is generally too long, and should be reorganized. The first paragraph was to show the significance of "SOC stabilization mechanisms and temperature sensitivity in terraced soils", and then What's the factors influenced SOC stabilization mechanisms and temperature sensitivity, … terraced soils affected can vary which factors…. aim of this study? ...what's your hypothesis…and finally add a sentence or two about the research expectations or significance in the end? Authors excessively cleared the well-known organic dynamic mechanisms; therefore, the relative information should be simplified.

**Reply**: We agree that the Introduction is a bit too long and need to be reorganized. We will rewrite the paragraph and the main revisions will be: 1) we will reduce the text about the well-known SOC stabilization mechanisms, e.g., introduction of SOC physical and chemical protection mechanisms (in Line 49-58); 2) We will reorganize the introduction as suggested by Referee. First paragraph will be a brief introduction of agricultural terraces and a short review about the current studies on SOC dynamics in terrace systems. At the end of this paragraph, we will point out the understudied question: understanding of SOC stabilization mechanisms in the agricultural terrace system. Second paragraph we will talk about two potential mechanisms (i.e., soil redistribution and SOC burial) that may affect SOC stabilization in terraces, and highlight the importance of mechanistic understanding of SOC stabilization mechanisms in assessing the role of terracing in terrestrial SOC cycle. Third paragraph we will talk about the significance of SOC temperature sensitivity study in evaluating the future warming-induced changes in terrace SOC stock. We will then review the potential controlling factors (e.g., SOC protection and quality) on SOC temperature sensitivity and formulate our research hypothesis: 'the terrace age is a fundamental driver of evolution of soil geochemical properties, which in turn determines the SOC protection and quality thus SOC stabilization and temperature sensitivity'. Last paragraph is about the significance of this research and specific research objectives.

Line 170-174, Delete the sentences "Soil C:N ratio... of the substrate (Soares & Rousk, 2019)", the information should not appear in the Material and Method section.

Reply: Thank you for the suggestion, this will be moved to the Discussion section in the revised draft

Discussion section is unnecessarily long and need thorough revision. The title of "4.1 Controls on SOC stability: importance of carbon burial and terrace age" and "4.2 Main controls on temperature sensitivity" are basically ok, but in each part, the logical thread of the discussion part is not clear.

Reply: Thanks for the comments. We agree that the Discussion section could be improved with a clearer structure. In doing so we will rework the Discussion section with a focus on (1) reducing the redundant contents that are not closely related to the main results (e.g., Line 380-390); (2) improving the logic and connection of each part of Discussion, especially in 4.1 section, we do identify this kind of issue.

---

## Author Response (AR1)

Dear editor and reviewers,

We thank editor for giving us the opportunity to submit the revised version of our manuscript. We appreciate the time and effort that reviewers have dedicated to improving the manuscript. We have checked the paper thoroughly and revised it according to the comments. Changes to be made to the manuscript will be explained here in response to each comment and have been added to the revised version of the manuscript. We hope all our efforts can make this paper suitable for publication.

**Comment from Associate Editor**

There are too many figures and i suggest reduce the numbers, e.g., put together Fig. 7 with Fig. 6a

Reply: Thanks for the comment. We have combined Fig. 6 with Fig. 7 and Fig. 3 with Fig. 4. As a result, we reduced number of figures from 9 to 7.

**Response to Anonymous Referee #2**

General Comments:

I have reviewed " SOC stabilization mechanisms and temperature sensitivity in old terraced soils." The study aimed to explore the stabilization mechanisms and temperature response of SOC in an old terraced soil". The topic is very interesting and within the scope of the journal. Generally, the manuscript is very well structured and presents the obtained results. The conclusion has highlighted the implication of terracing on soil carbon stabilization in the context of climate warming. However, below are some corrections:

Reply: We thank the reviewer for this positive evaluation and constructive comments, which have been very helpful to improve the quality of the manuscript.

Lines, 15-20. Our mechanistic understanding of soil organic carbon (SOC) (de-) stabilization mechanisms and of the persistence of SOC stored in terraced soils, however, is far from complete. Here we explored the factors controlling 20 SOC stability and temperature sensitivity (Q10) of abandoned prehistoric agricultural terrace soils in NE England, using soil fractionation and temperature sensitive incubation in combination with measurements of terrace soil burial age. Should be replaced with "However, our mechanistic understanding of soil organic carbon (SOC) (de-) stabilization mechanisms and the persistence of SOC stored in terraced soils is far from complete. Here we explored the factors controlling SOC stability and temperature sensitivity (Q10) of abandoned prehistoric agricultural terrace soils in NE England, using soil fractionation and temperature sensitive incubation combined with terrace soil burial age measurements."

Reply: Thanks for the comment. We have revised the sentences accordingly.

**Line 18-21** now reads 'However, our mechanistic understanding of soil organic carbon (SOC) (de-) stabilization mechanisms and the persistence of SOC stored in terraced soils is far from complete. Here we explored the factors controlling SOC stability and temperature sensitivity ($Q_{10}$) of abandoned prehistoric agricultural terrace soils in NE England, using soil fractionation and temperature sensitive incubation combined with terrace soil burial age measurements'

Line 20 suggest that burial should be "suggest that the burial" Line 55 function of molecular complexity should be "function of the molecular complexity, through cut should be "through the cut." Line 70 Both the current SOC status should be "The current SOC status"

**Reply**: Thanks for the comment. We have revised the sentences (see **Line 24, 48**).

Lines 159 that represent should be replaced with "representing" Lines 255 between the terrace soil layers and control soil layers should be "between the terrace and control soil layers"

**Reply**: We revised the text accordingly.

**Line 137-138** now reads 'Forty gram of 5 mm sieved air-dried soils were fractionated in duplicate to obtain SOC fractions for selected soil layers that representing different layers within each undisturbed and terrace soil'

Lines 320 relative should be changed to "relatively." Lines 335 In order to further should be "To further"

**Reply**: Agreed, this has been revised in the manuscript.

**Line 272** now reads 'Linear regression further showed that for relatively younger soil horizons, …'

**Line 277** now reads 'To further identify the underlying controls, we linked the patterns of $Q_{10}$ to soil C:N ratio and measured soil geochemical properties'

Lines 360 have a significant lower should be "have a significantly lower."

**Reply**: Revised (see Line 298)

Lines 370 non-terraced should be corrected "non-terraced."

**Reply**: Thanks for the comment. We corrected the sentence (see Line 312).

Lines 380 with the observation should be changed to "by observing."

**Reply**: We agree and have corrected this.

Lines 390 require should be "requires" Lines 395 which can dampen, consider changed to "dampening."

**Reply**: We agree, revised (Line 326, Line 329).

**Line 328-329** now reads 'However, environmental constraints (e.g., SOC physical or chemical protection, soil acidity etc.) can reduce substrate availability, dampening the intrinsic temperature sensitivity'

Lines 400 relative should be changed to "relatively."

**Reply**: Thanks for the comment. We corrected this throughout the whole manuscript (Line 333).

**Response to Anonymous Referee #2**

We gratefully thank the referee for his/her constructive comments and have revised the manuscript accordingly. In our response below, referee comments are shown in black, our response in blue.

General Comments:

This manuscript explored the controlling factors for SOC stability and temperature sensitivity of its decomposition in agricultural terrace soils. The result is interesting and has a value for evaluate SOC stability under man-made landform. The manuscript falls into the scope of BG, although the Introduction and Discussion sections need to be improved.

Reply: We thank the reviewer for these constructive comments. We revised the Introduction and Discussion accordingly and are confident that the presentation is now easier to follow and understand. Details are as follows.

Introduction is generally too long, and should be reorganized. The first paragraph was to show the significance of "SOC stabilization mechanisms and temperature sensitivity in terraced soils", and then What's the factors influenced SOC stabilization mechanisms and temperature sensitivity, … terraced soils affected can vary which factors…. aim of this study? …what's your hypothesis…and finally add a sentence or two about the research expectations or significance in the end? Authors excessively cleared the well-known organic dynamic mechanisms; therefore, the relative information should be simplified.

Reply: In response to this comment, we have revised and shortened the introduction. We revised Introduction section with focus on: 1) we reduced the text about the well-known SOC stabilization mechanisms, e.g., introduction of SOC physical and chemical protection mechanisms; 2) We reorganized the introduction as suggested by Referee #2. First paragraph is a brief introduction of agricultural terraces and a short review about the current studies on SOC dynamics in terrace systems. In the second paragraph we discussed three potential mechanisms that may affect SOC stabilization in terrace systems, and highlighted the importance of mechanistic understanding of SOC stabilization mechanisms in assessing the role of terracing in terrestrial SOC cycle. In the third paragraph we addressed the significance of SOC temperature sensitivity for evaluating future warming-induced changes in terrace SOC stocks. We then reviewed the potential controlling factors (e.g., SOC protection and quality) on SOC temperature sensitivity and formulate our research hypothesis: H1 'age since burial is a fundamental factor driving the evolution of soil geochemical properties in terraced soils', H2 this co-evolution controls SOC protection and quality and thus SOC stabilization and temperature sensitivity'. The last paragraph dealt with the significance of this research and specific research objectives. Please check the revision from **Line 35-85**).

Line 170-174, Delete the sentences "Soil C:N ratio… of the substrate (Soares & Rousk, 2019)", the information should not appear in the Material and Method section.

Reply: Thanks for the comments. We have deleted these sentences.

Discussion section is unnecessarily long and need thorough revision. The title of "4.1 Controls on SOC stability: importance of carbon burial and terrace age" and "4.2 Main controls on temperature sensitivity" are basically ok, but in each part, the logical thread of the discussion part is not clear.

Reply: Thanks for the comments. We agree that the Discussion section could be improved with a clearer structure. In doing so we rework the Discussion section with a focus on (1) reducing the redundant contents that are not closely related to the main results (e.g., Line 382-389 in preprint); (2) improving the logic and connection of each part of Discussion, especially in 4.1 section, we do

identify this kind of issue. (Line 295-322)